# Creating Value in Non-Profit Sports Organizations: An Analysis of the DART Model and Its Performance Implications

**Jorge Iván Brand Ortíz [1], Silvana Janeth Correa Henao [1], Laura Cristina Henao Colorado [1] and Alejandro Valencia-Arias [2,*]**

[1] Ciencias Administrativas, Instituto Tecnológico Metropolitano, Medellín 050004, Colombia; jorgebrand@itm.edu.co (J.I.B.O.); silvanacorreaitm@gmail.com (S.J.C.H.); laurahenao@itm.edu.co (L.C.H.C.)

[2] Escuela de Ingeniería Industrial, Universidad Señor de Sipán, Chiclayo 14001, Peru

[*] Correspondence: valenciajho@crece.uss.edu.pe; Tel.:+51-573002567977

**Abstract:** This study aims to analyze the impact of value creation and cocreation, as measured by the dialogue, access, risk assessment, and transparency (DART) model, on the performance of non-profit sports organizations (NPSOs). To assess this impact, the authors analyzed data collected from sports and administration managers of NPSOs, specifically Colombian amateur soccer clubs. The study used partial-least-squares structural-equation modeling (PLS-SEM) with constructs of the reflexive-formative type. The results indicate that value creation has a positive impact on the performance of NPSOs through the mediating effect of value cocreation. Despite the limitations of this study, including the limited research on the relationship between value creation and cocreation and NPSO performance in Colombia, the findings contribute to the understanding of the mediating effect of cocreation. The authors found that cocreation mainly affects the sport, customers/members, communication and image, finance, and organization dimensions of NPSOs in developing countries. This study draws attention to the potential benefits of cocreation for NPSOs and emphasizes the importance of creating value in this context. The study concludes that further studies on the constructs proposed in this research would help to understand the phenomenon of innovation and its impact on NPSOs. Overall, this study provides valuable insights for managers and policymakers in NPSOs—especially in developing countries—on the importance of value cocreation in improving their performance.

**Keywords:** NPSOs; value creation; value cocreation; performance of sports organizations; amateur soccer clubs

## 1. Introduction

Innovation in the management of sports organizations is a topic of great interest because it enhances the performance of organizations and increases their competitive advantage [1,2]. One example is that of sports clubs, which have improved their efficiency and effectiveness through innovation [3,4], thereby gaining a competitive advantage [5].

The term value creation also involves innovation, which increases customer-perceived benefits [6,7]. Flexibility and innovation are critical to organizational performance and goal achievement because they optimize processes and maximize profits [8]. Value creation in sports goods and/or services has emerged as an opportunity to foster innovation and take advantage of market opportunities. Some studies have concluded that increasing value creation in sports organizations, such as soccer clubs, can positively affect their finances [5,9]. For example, Udinese Calcio, an Italian soccer club, created value through the training and transfer of players, thus obtaining positive economic results [10]. Furthermore, a study on English soccer clubs established that the development of talented athletes and the retaining of experienced players affects the financial performance of professional clubs [11]. Furthermore, the COVID-19 pandemic highlighted the importance of creating value in



sports entities during times of crisis because consumers require solutions and innovations in real time [12].

Several studies agree that innovation through value creation has cocreation as its present and future foundation [13–16]. In particular, the new approach to the creation of value through interaction between organizations and stakeholder groups is known as value cocreation [17]. Organizations play an increasingly important role in the design of value-creation strategies based on collaboration between sports organizations and customers. For example, in sporting events, cocreation occurs during the interactions between athletes, fans, coaches, and staff [18]. A study on the benefits of cocreation in sports-organization settings analyzed mega-events, such as the FIFA World Cup and the Olympic Games, and identified different forms of association between stakeholder groups [19]. Another study on value cocreation in Fan Fests focused on the relationship between sports experience and fan consumption [20].

In Colombia, research on sport management is conducted independently, that is, it does not follow a rigorous and valid information system, which makes comparisons difficult. Moreover, research on value creation and cocreation in sports clubs is nonexistent. In particular, soccer clubs in the country are faced with interventions in their organizational dynamics, scarce financial resources, a lack of adequate sports facilities, attrition, and an increasingly competitive environment. All these conditions lead to insufficient value creation and prevent sports organizations from improving their performance and, therefore, their competitive advantage [21,22]. In contrast, some authors have highlighted the importance of conducting studies on sports organizations by taking into account their strengths [23,24]. For example, the literature explains that sports clubs that have their own venues obtain greater economic benefits because they can carry out other projects there.

Consequently, this study aims to measure and understand the impact of value creation and cocreation on the performance of amateur soccer clubs in Colombia.

## 2. Literature Review and Hypothesis

### 2.1. Value Creation and Cocreation

There is no widely accepted definition of value creation. However, it can be understood as new or adapted products, services, or activities that are valued and accepted by knowledgeable consumers or users as creative acts in a specific context [25]. To understand the impact of value creation, we should consider where, how, and when value is created. In addition, value is sometimes cocreated. Value cocreation, in turn, is an activity that increases the value of products, services, or actions through the collaboration of the different actors or parties involved [26,27].

In sports management, fans have been found to engage in a series of behaviors that can benefit both the sports entity and other fans. For example, attendees at live sporting events socially interact with other spectators, which can enhance their experience of the game. Fans also join organizations related to their sport or team, such as fan clubs and alumni associations [28]. In fact, in the digital age, fans are more active and exhibit online behaviors such as accessing teams' social media channels while watching live sports [29].

Some authors suggest that sports organizations should consider the expectations and motivations of different customer groups and provide offerings designed to meet the specific needs of different fan segments based on the spectators' experiences with the sports product and the distance traveled to attend the sporting event [30]. Sporting-event planners may be interested in the creation of value through collaborative spectators' characteristics, such as their level of knowledge, fairness, interaction, personality, and relationships, as it is thought that spectators are more likely to endorse these secondary characteristics as the degree of collaboration to create value increases [31]. Further studies identified two types of spectator: known and unknown [32]. These studies examined the effects of interactions within a framework based on the dominant-customer logic and the sports value, incorporating elements such as on-field sports performance, off-field service quality, overall satisfaction, and team identification.

The sports industry is experiencing a steady boom. Thanks to its cultural and symbolic nature and the involvement of its stakeholders, it provides a dynamic environment for research. In this context, there is a shift in the way in which value is created. Fans are no longer mere passive recipients of value, but they can play an active role as value cocreators [28].

Cocreation in sports has influenced the traditional concept of value creation [33,34]. Studies have demonstrated that sports organizations have been affected in the way in which they conceive sporting-event consumption [19], in the form of increasing fans' consumption intention, and in the importance they give to knowing consumers' value dimensions when offering sports services [35].

Other authors have examined different value-cocreation platforms, such as Fan Fests, which are spaces where sports customers interact to cocreate value [20]. However, the impact of value cocreation on the performance dimensions of sports organizations has not been studied in depth [32]. To gain competitive advantages and increase market share, event management should go beyond improving internal efficiency. Instead, the goal should be to facilitate collaborative efforts to create value, considering that studies have primarily focused on developing frameworks to examine spectator-sports tourism, analyzing value cocreation in gyms, or exploring consumer-behavior issues related to spectator sports [33].

### 2.2. Performance of Sports Organizations

Performance management aims to optimize the efficiency and effectiveness of an organization by measuring and evaluating the use of its resources [15]. These two concepts are directly related to the achievement of goals through the use and allocation of resources; nevertheless, they are different. Effectiveness focuses on developing strategies to maximize the expected results; efficiency focuses on minimizing the use of resources according to the goal to be achieved. However, both efficiency and effectiveness are parts of organizational performance [36–39].

Although there are several studies on the performance of sports organizations, most of them focus on for-profit organizations, and only a few focus on non-profit organizations. Typically, performance management in sports organizations is based on the following elements: (I) clear and precise objectives; (II) performance indicators that match these objectives; (III) adequate management to achieve the objectives; (IV) the proper measurement of the selected indicators; and (V) constant reviews of the progress to provide feedback on objectives, indicators, goals, and actions [40].

The studies that analyze the performance of sports organizations mainly focus on national governing bodies or professional sports clubs, leaving aside amateur sports institutions in developing countries [15]. For this reason, it is imperative to conduct studies on these types of organizations, such as amateur soccer clubs in Colombia. In addition, it is clear from the literature that the measurement of organizational performance involves multiple dimensions, including creativity, innovation, productivity, effectiveness, efficiency, competitiveness, and profitability [41]. Similarly, the measurement of performance in non-profit sports organizations (NPSOs) requires multiple criteria [42].

Due to the different perspectives on organizational performance in the literature, there is a lack of consistency in performance measurement [43]. However, several studies have conceptualized performance measurement in NPSOs [42]. In one of the few focused on identifying the dimensions for measuring performance in NPSOs based on the adoption of several models [44], the authors found that performance in sports organizations can be measured in five dimensions (Figure 1).

### 2.3. Hypotheses and Conceptual Model

Organizations in general need to create value to differentiate their products, services, or activities and consolidate them in the long term [9]. This has led sports organizations to introduce innovations to enhance their performance and take advantage of market

opportunities [5,45]. Increased value creation has been proven to have an impact on the economic benefits received by sports organizations [9].

**Figure 1.** Conceptual model.

Different studies in this field have implicitly shown how the performance of sports organizations is directly influenced by the implementation of value-creation strategies [10,11,46]. Furthermore, NPSOs have been shown to compete to increase efficiency and effectiveness through financial support, sports results, and member participation in their programs [44].

Therefore, the relationship between value creation and value cocreation is currently a topic of interest in the literature on management because it has been stated that value originates in the interaction between an organization and its customers—and it is the latter who create value [47]. However, it is only when there is a joint interaction that cocreation experiences occur [48]. In line with these observations, some authors argue that value cocreation occurs when customer value creation is transferred within the organization through collaborative relationships beyond the commercial domain and through communication using the different channels provided by marketing strategies [49]. This leads to knowledge exchange, skill acquisition, and organizational learning in the value-creation process.

Thus, depending on how value creation is defined, the concept of value cocreation can have different meanings. Value creation becomes a structured process in which companies and customers have well-defined roles and objectives, while value cocreation refers to

situations in which organizations invite customers to participate in their various processes, and customers agree [50,51]. This relationship between value creation and cocreation is the new dominant logic [52], in which value creation is not understood as produced by an organization but rather as created in a collaborative process between parties. Hence, sports organizations embrace value cocreation as a way of creating value in a dynamic and collaborative manner [30,53].

Furthermore, value creation seeks to offer new and better products, services, or activities to customers, which, in turn, leads to greater benefits for organizations [54]. Moreover, it involves improving the performance and competitive advantage of organizations [6,7]. According to the theory of value creation, organizations need to differentiate their products, services, or activities to be sustainable in the long term [9]. Consequently, they need to take advantage of market opportunities to improve their performance, which results in increased value creation. In addition, following the new customer-oriented logic, it is customers who create value, which influences the finances of sports organizations [2,5,9,10,43].

In contrast, some authors state that performance in sports organizations, particularly in non-profit organizations, has a different meaning, although it is also focused on value creation [44]. In other words, NPSOs do not compete for profits but aim to increase efficiency and effectiveness to obtain benefits [54], such as financial support, sports results, and increased member participation in the programs they offer. Consequently, it is imperative for them to create value to positively affect their income, financial results, sports outcomes, reputation, and relationship through communication and image. Similarly, it has been suggested that value creation in soccer organizations should be evaluated by taking into account the dimensions of income, sport, education in values, and communication [9].

Finally, as explained above, the creation of value through interactions between organizations and their stakeholders is called value cocreation [9]. Thus, value creation is strengthened and obtains greater benefits for organizations through the collaboration of multiple stakeholders who contribute knowledge, experience, and skills to cocreate products, services, or activities and enhance the organization's performance [9,55,56]. Since 15 years ago, when the inclusion of collaborative networks, customers, resources, services, and the relationships between them began to be studied, different studies have explained that there is a relationship between value cocreation and organizational performance [57–59].

Based on the above and the theoretical foundations, we formulate the following hypotheses.

H1. Value creation directly and positively influences value cocreation between sports organizations and their customers. H1a: Higher levels of value creation lead to higher levels of sports performance in sports organizations. H1b: Higher levels of value creation lead to higher levels of customer/member performance in sports organizations. H1c: Higher levels of value creation lead to higher levels of communication and image performance in sports organizations. H1d: Higher levels of value creation lead to higher levels of financial performance in sports organizations. H1e: Higher levels of value creation lead to higher levels of organizational performance in sports organizations. H2a: Value cocreation positively mediates the relationship between value creation and sports performance in sports organizations. H2b: Value cocreation positively mediates the relationship between value creation and customer/member performance in sports organizations. H2c: Value cocreation positively mediates the relationship between value creation and image and communication performance in sports organizations. H2d: Value cocreation positively mediates the relationship between value creation and financial performance in sports organizations. H2e: Value cocreation positively mediates the relationship between value creation and organizational performance in sports organizations.

Finally, the conceptual model applied in this study relates value creation and cocreation to the performance dimensions of sports organizations (Figure 1).

We used partial-least-squares structural-equation modeling (PLS-SEM) because the conceptual model has many items (54), second-order constructs (2), and dimensions (13), mainly because of the presence of formative and reflective second-order constructs in

the model [60,61]. To analyze the data using PLS-SEM, we employed SmartPLS software (v. 3.2.7.). To measure the model and its structure, we relied on [62].

## 3. Research Methodology

### 3.1. Research Context

For this study, we selected Colombian amateur soccer clubs. The selected clubs are part of the Liga Antioqueña de Fútbol (LAF). The LAF is considered the most important amateur soccer league in Colombia thanks to its administrative and sports results. Currently, 650 clubs and 22,140 soccer players participate in the tournaments organized by the LAF.

### 3.2. Dimension Definition and Operationalization

Value creation, as a source of competitiveness, is based on convergence of ideas, collaborative agreements, and cocreation of experiences with stakeholders, which originate in different internal and external sources [43]. For organizations, relationships with networks of partners and customers are important to develop new skills, discover new technologies, become familiar with new processes and structures, and establish new partnerships for mutual benefit [62], based on the dimensions of interest in value-creation measurement.

In this regard, the author of [63] conducted a literature review to systematically develop a validated scale for measuring business-model innovation, with value creation as one of its main dimensions. The author selected common components of different value-creation models, thus obtaining 33 unique components that were divided as follows: (1) new skills, (2) new technologies/equipment, (3) new processes and structures, and (4) new partnerships.

With respect to value cocreation—which has been defined as a new source of competitive advantage for sports organizations—and following Prahalad and Ramaswamy [9,46], we adopted the DART model, which offers four dimensions for measuring value cocreation: (1) dialogue, through which knowledge is shared among stakeholders; (2) access, which leads the organization to exchange information on value and, thus, create positive experiences; (3) risk assessment, in which more information and responsibility are required for value creators to manage the risks involved in cocreated goods; and (4) transparency, which is essential for reducing information asymmetry and building the trust necessary for interactions between organizations and customers.

In addition, as noted above, the performance of sports organizations has not been studied in depth. Studies on other sectors have provided performance data from different theoretical perspectives, without a clear consensus [45,47]. As a result, there is no single commonly accepted definition of organizational performance in the literature on management [41]. However, the existing definitions are mainly based on generally accepted organizational performance models, which address this factor through dimensions including goal achievement, available system resources, internal processes, strategic groups, and competitive values [40,45].

Consequently, for the purpose of this study, we employed the three main models proposed by [64] to measure the performance of sports organizations: system-resource model, internal-process model, and goal-achievement model. It should be noted that the organizational performance construct in the sports sector is based on common components found in the proposals in studies such as [62,65,66], where it was addressed from different perspectives. The latter two studies are particularly relevant to the sports sector and NPSOs.

Therefore, according the proposed model, the organizational performance factor is measured through the following dimensions: (1) elite sport, including international sports results and participation of athletes in international competitions; (2) customers, specifically offerings to non-competitive customers who require sport services; (3) communication and image, involving control of the external environment that is responsible for promoting sport and communication for its members and clubs; (4) finance, comprising management of financial resources for the survival of sports organizations; and (5) organizational dimension, encompassing qualification of human talent for the operation of sports organizations.

Based on the information above and our understanding of the nature of the proposed structural model, the measurement model consists of both formative and reflective constructs. The formative construct value creation is composed of four dimensions, with a total of 13 measurement variables: new skills (3 variables), new technology/equipment (3 variables), new partners (4 variables), and new processes (3 variables) [63]. Similarly, the construct of value cocreation consists of four dimensions taken from the DART model, which, in turn, comprise 21 measurement variables: dialogue (6 variables), access (5 variables), risk (5 variables), and transparency (5 variables) [67]. Furthermore, the reflective-construct measurement model comprises five dimensions that reflect the performance of sports organizations. These five dimensions cover a total of 20 measurement variables: sport (4 variables), customers/members (4 variables), communication and image (4 variables), finance (4 variables), and organization (4 variables) [62]. Due to the context of this study, the dimensions grassroots and elite sport, proposed in [62], were replaced by sport (see Appendix A—Table A1). To measure the items proposed in the study, we used a Likert scale, where 5 = strongly agree and 1 = strongly disagree.

### 3.3. Instrument

To collect data, we used a questionnaire based on the 54 measurement variables of the study. All items were translated into Spanish and adapted to the specific context of Colombian amateur soccer clubs. To ensure the reliability and initial validity of the instrument [48], we performed two preliminary tests with executives from the Colombian sports sector, as well as a pilot test with sports and administration managers of Colombian amateur soccer clubs.

### 3.4. Procedure

The self-administered questionnaire was sent to sports and administration managers of amateur soccer clubs between November and December 2018. The participants received the questionnaire along with an informed consent form. A total of 322 managers accepted the invitation to respond to the questionnaire. After receiving the questionnaires completed, 27 (8%) were excluded due to missing information exceeding 15% of the total [61], which resulted in a valid sample of 305 questionnaires. Table 1 shows the demographic characteristics of the study participants.

**Table 1.** Demographic characteristics of the participants.

|  |  | N | Percentage (%) |
|---|---|---|---|
| Sex | Male | 288 | 94 |
|  | Female | 17 | 6 |
| Education | Undergraduate degree | 153 | 50 |
|  | High School | 111 | 36 |
|  | Postgraduate degree | 41 | 14 |
| Position | Middle manager | 126 | 40 |
|  | Senior manager | 97 | 31 |
|  | Coordinator | 82 | 29 |

We used convenience sampling because we needed to invite only soccer clubs that were members of the LAF. In addition, the questionnaire was administered at a specific time and place, that is, during a convention attended by clubs from all over the country. Only one representative (i.e., a high-ranking administrative officer) from each club responded to the questionnaire to avoid duplicate responses.

### 4. Statistical Analyses

#### 4.1. Measurement Models

In the conceptual model, value cocreation and value creation are considered unidentifiable second-order constructs because they are formed by their measurement variables

but not reflected in them. To solve this problem, we implemented the two-step build-up approach proposed in [61]. The measurement variables of the formative constructs (regardless of whether they were unidentifiable or reflective) were related to the other constructs in the model. Subsequently, the results or latent variables were considered as variables reflected in these constructs for identification purposes.

Table 2 presents acceptable Cronbach's Alpha (CA) and composite reliability (CR) values for all the dimensions, thus demonstrating the internal consistency and reliability of the instrument. In addition, the average variance extracted (AVE) values were found to be greater than 0.5, and the load sizes were greater than 0.6 and statistically significant. These values suggest satisfactory convergent validity and demonstrate the robustness of the items in measuring the respective dimensions.

**Table 2.** Measurement model of the reflective constructs (internal consistency, reliability, AVE, and coefficient).

| Dimensions | Items | Loadings * | CA | CR | AVE |
|---|---|---|---|---|---|
| Sport | Sport_1 | 0.767 *** | 0.880 | 0.918 | 0.737 |
| | Sport_2 | 0.912 *** | | | |
| | Sport_3 | 0.894 *** | | | |
| | Sport_4 | 0.852 *** | | | |
| Customers/members | Customer_M_1 | 0.824 *** | 0.828 | 0.886 | 0.661 |
| | Customer_M_2 | 0.736 *** | | | |
| | Customer_M_3 | 0.838 *** | | | |
| | Customer_M_4 | 0.849 *** | | | |
| Communication and image | Com_Image_1 | 0.800 *** | 0.885 | 0.921 | 0.744 |
| | Com_Image_2 | 0.867 *** | | | |
| | Com_Image_3 | 0.911 *** | | | |
| | Com_Image_4 | 0.868*** | | | |
| Finance | Finance_1 | 0.761 *** | 0.855 | 0.902 | 0.699 |
| | Finance_2 | 0.887 *** | | | |
| | Finance_3 | 0.869 *** | | | |
| | Finance_4 | 0.820 *** | | | |
| Organization | Organization_1 | 0.883 *** | 0.909 | 0.936 | 0.786 |
| | Organization_2 | 0.890 *** | | | |
| | Organization_3 | 0.910 *** | | | |
| | Organization_4 | 0.863 *** | | | |

* Indicates significant paths: *** $p < 0.001$.

Discriminant validity was demonstrated according to the criterion proposed in [53]. All the reflective dimensions met the criteria because the square root of each AVE value was greater than the correlations between the dimensions presented below the diagonal, as shown in Table 3.

**Table 3.** Measurement model of the reflective constructs (discriminant validity).

| | 1 | 2 | 3 | 4 | 5 |
|---|---|---|---|---|---|
| Communication and image (1) | 0.863 | 0.946 | 0.773 | 0.802 | 0.884 |
| Customers/Members (2) | 0.813 | 0.813 | 0.808 | 0.818 | 0.924 |
| Finance (3) | 0.679 | 0.680 | 0.836 | 0.797 | 0.753 |
| Organization (4) | 0.720 | 0.709 | 0.705 | 0.887 | 0.792 |
| Sport (5) | 0.787 | 0.796 | 0.662 | 0.716 | 0.858 |

The discriminant validity of most factors was demonstrated, except for the following relationships: Communication and image—customers/members, communication and image–Sport, and customers/members—sport (gray cells in Table 3). Since these values are very close to the 0.9 criterion, HTMT inference was verified by running complete bootstrapping, producing a 95% confidence interval for the HTMT between the constructs

in question. We found that the interval did not include 1 [52]. Therefore, the communication and image, customers/members, and sport constructs were different, and discriminant validity was assumed. In addition, the content validity of the measurement scales was also demonstrated because they met the Fornell–Larcker and HTMT inference criteria (very close to the limit). Regarding the other relationships, the HTMT inference criterion was also met.

The weight–load ratio of the indicator and its significance for the formative dimensions of value cocreation and value creation was demonstrated [61]. Table 4 shows that all the weights of the dimensions were significant ($p > 0.001$). The variance-inflation factor (VIF) for the formative dimensions was also evaluated. The VIF and tolerance (TOL) values, which are presented in Table 4, were accepted. This means that the formative dimensions were not correlated.

**Table 4.** Measurement model of the formative constructs.

| Second-Order Construct | Dimensions | Collinearity Statistics | | Weight-Load |
|---|---|---|---|---|
| | | TOL | VIF | Sig. Weight * |
| Value cocreation | Dialogue | 0.80 | 1.247 | Yes |
| | Access | 0.82 | 1.205 | Yes |
| | Risk | 0.76 | 1.300 | Yes |
| | Transparency | 0.78 | 1.275 | Yes |
| Value creation | New skills | 0.79 | 1.257 | Yes |
| | New technology | 0.82 | 1.211 | Yes |
| | New partners | 0.82 | 1.216 | Yes |
| | New processes | 0.76 | 1.301 | Yes |

\* Indicates significant paths.

### 4.2. Structural-Model Analysis

In this study, the structural model was evaluated in three steps [61]: the coefficient of determination ($R^2$), predictive relevance ($Q^2$), and path coefficients of the structural model. To this end, we employed SmartPLS software. The $R^2$ values for the endogenous dimensions (i.e., sport, customers/members, communication and image, finance, and organization) were above the level of 10% recommended in [64]. Following the general rules, the $R^2$ values of customers/members (0.27), organization (0.26), communication and image (0.25), sport (0.25), and finance (0.24) were weak. Based on the Blindfolding function, all the $Q^2$ values were above zero: organization (0.193), communication and image (0.177), customers/members (0.170), sport (0.170), and finance (0.158). These values suggest the predictive relevance of the model regarding the endogenous reflective dimensions.

Finally, to analyze the path coefficients of the structural model, a bootstrap method with 500 random samples with replacement was employed [51]. Table 5 shows that there was a direct relationship between value creation and sport (β = 0.33 \*\*\*), customers/members (β = 0.34 \*\*\*\*), communication and image (β = 0.29 \*\*\*), finance (β = 0.27 \*\*\*\*), and organization (β = 0.28 \*\*\*\*). These results demonstrate that value creation has a significant impact on the performance dimensions of amateur soccer clubs in Colombia, supporting H1a, H1b, H1c, H1d, and H1e.

### 4.3. Test for Mediation

The study used the criteria suggested in [61] to carry out the measurement: (i) bootstrapping to estimate the coefficients; (ii) the calculation of the product of the coefficients; (iii) the calculation of the standard error of the estimate; (iv) the calculation of the significance by dividing the indirect effect by the standard error [7]; and (v) the calculation the variance accounted for (VAF) by dividing the indirect effect by the total effect. Table 6 shows that all the mediations were significant. In addition to the agreement with the VAF, there was full mediation between value creation and the dimensions of customers/members,

communication and image, finance, and organization through value cocreation. There was also partial mediation between value creation and sport through value cocreation.

**Table 5.** Significant testing results of the structural model path coefficients.

| Structural Path | Path Coefficient | *t*-Value | Conclusion |
|---|---|---|---|
| Value creation → Sport | 0.339 *** | 6.870 | H1a: Supported |
| Value creation → Customers/members | 0.340 *** | 6.867 | H1b: Supported |
| Value creation → Communication and image | 0.294 *** | 6.144 | H1c: Supported |
| Value creation → Finance | 0.278 *** | 6.978 | H1d: Supported |
| Value creation → Organization | 0.280 *** | 6.6693 | H1e: Supported |
| Value creation → Value cocreation | 0.575 *** | 15.260 | Supported |
| Value cocreation → Sport | 0.220 *** | 4.012 | Supported |
| Value cocreation → Customers/members | 0.250 *** | 5.546 | Supported |
| Value cocreation → Communication and image | 0.276 *** | 6.104 | Supported |
| Value cocreation → Finance | 0.281 *** | 6.674 | Supported |
| Value cocreation → Organization | 0.297 *** | 7.102 | Supported |

*** $p < 0.001$.

**Table 6.** Tests for mediation.

| Effect of | * Indirect Effect (*t*-Value) | Total Effect | VAF (%) | Interpretation | Conclusion |
|---|---|---|---|---|---|
| VC → VCC → Sport | 0.127 *** (3.76) | 0.161 | 0.79 | Partial mediation | H2a: Supported |
| VC → VCC → Customer_M | 0.144 *** (5.25) | 0.171 | 0.84 | Full mediation | H2b: Supported |
| VC → VCC → Com_Image | 0.159 *** (5.66) | 0.187 | 0.85 | Full mediation | H2c: Supported |
| VC → VCC → Finance | 0.162 *** (5.89) | 0.189 | 0.85 | Full mediation | H2d: Supported |
| VC → VCC → Organization | 0.171 *** (6.02) | 0.199 | 0.86 | Full mediation | H2e: Supported |

* Indicates significant paths: *** $p < 0.001$. Note: value creation (VC); value cocreation (VCC). VAF > 80% indicates full mediation, $20\% \leq$ VAF $\leq 80\%$ shows partial mediation, while VAF < 20% indicates no mediation.

## 5. Results and Discussion

The results of this study contribute to the understanding of the impact of innovation on the performance of NPSOs. The nomological validity of the conceptual model (see Figure 2) and the statistical results of the PLS-SEM prove that high levels of value creation lead to high levels of performance in Colombian amateur soccer clubs. Furthermore, the performance of the clubs is positively affected when it is mediated by value cocreation.

According to the results described above, the dimensions that explain the organizational performance of non-profit sports organizations are sport, customers/members, communication and image, finance, and organization. Specifically, sport and customers/members have the strongest influence on performance ($\beta = 0.340$ and $\beta = 0.339$, respectively). In addition, value creation and cocreation have a direct impact on organizational performance, which is stronger in two dimensions: organization and finance (VC → finance $\beta = 0.278$ and VC -> organization $\beta = 0.280$; VCC -> organization $\beta = 0.297$ and CCV -> finance $\beta = 0.281$). However, value cocreation has the greatest influence on organization and finance, while value creation has a significant effect on value cocreation ($\beta = 0.575$). This indicates that there is full mediation of the value cocreation between the impact of value creation and organizational performance, specifically in the dimensions contained in Table 6.

These results show similarities with those described in [55], where value creation was found to positively affect the performance of organizations, helping them to maintain their competitive advantage. The findings also have similarities with the conceptualizations found in [10,11,46], which reveal the importance that sports organizations attach to soccer to generate economic benefits through value creation. This study also confirmed that NPSOs need value creation to achieve innovations that improve their organizational performance [4,5,45].

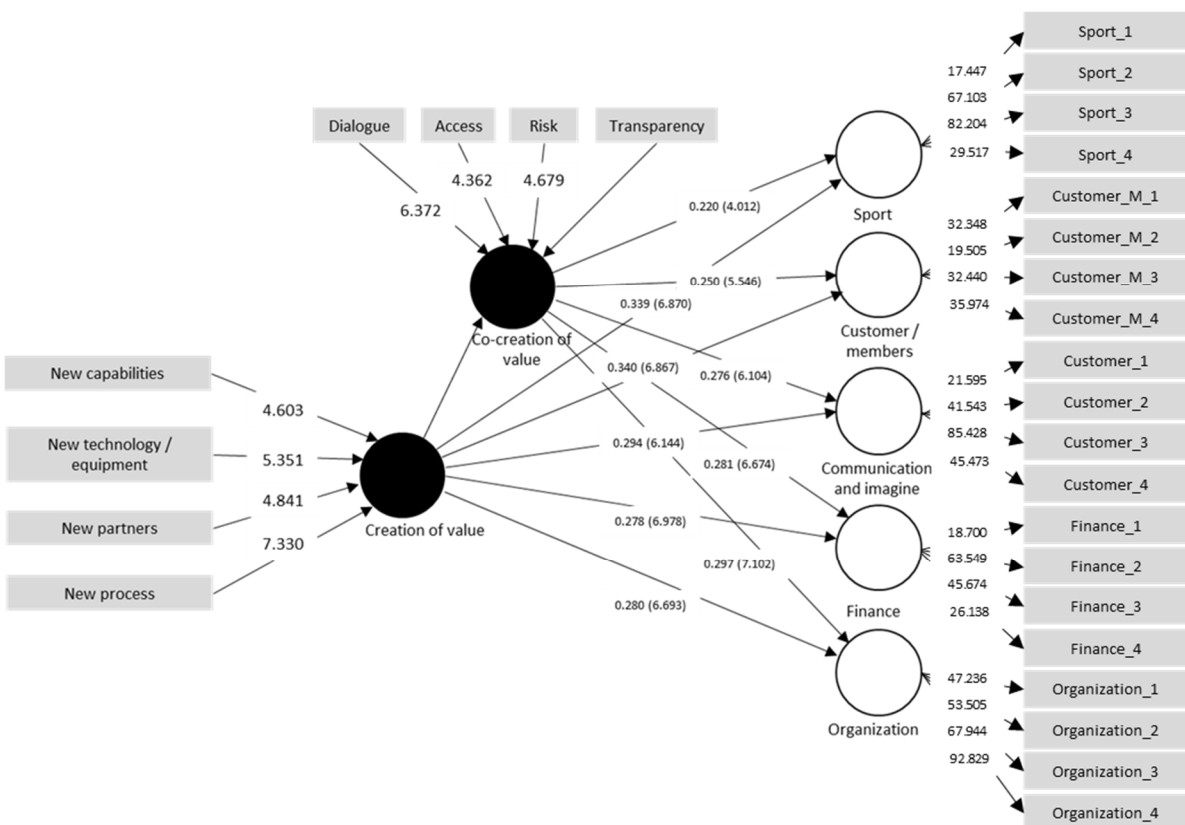

**Figure 2.** Results of the conceptual model.

Furthermore, the study showed that the dimension customers/members has the greatest positive impact on value-creation processes in Colombian amateur soccer clubs (see Table 5). This means that clubs can meet the needs of their customers/members with new or improved offerings [56]. Furthermore, clubs are able to attract new customers/members or a segment of the market by creating value in their products, services, or activities [68]. Moreover, clubs can develop and establish relationships with customers/members to ensure loyalty when their products and services are substitutable or create links that ensure future sales [64].

This is consistent with the findings described in [69], demonstrating the direct relationship between value creation for customers and organizational profits. Furthermore, according to the results of this study, clubs innovate by creating value, which positively affects services, loyalty, and attraction of new customers/members. Moreover, NPSOs compete for financial support, customer/member participation in offered programs, and sports results [44]. Additionally, the sport dimension, which has the second highest positive impact due to value creation (see Table 5), indicates that clubs can achieve sports results through value creation. This is consistent with the findings reported in [11].

In terms of mediation relationships, this study revealed that value creation through value cocreation positively affects the performance of Colombian amateur soccer clubs (see VAF in Table 6). Thus, the mediating effect of the DART model is full between value creation and customers/members (VAF = 84%), communication and image (VAF = 85%), finance (VAF = 86%), and organization (VAF = 86%). These results indicate that creating value in a dynamic and collaborative way stimulates value cocreation [33,34].

### 5.1. Implications

This study aims to close an important research gap in the operation of NPSOs:. Previously, the impact of value creation and cocreation in these organizations had not been explored in depth. However, understanding how value creation and cocreation affect the

success, sustainability, and growth of NPSOs is crucial. To bridge this gap, the present study employed the DART conceptual model and the PLS-SEM method to analyze the dimensions that explain organizational performance in the context of Colombian amateur soccer clubs. By revealing the positive relationship between high levels of value creation and NPSO performance, along with the mediating effects of value cocreation, this study helps to fill the knowledge gap in this area. Furthermore, it offers valuable insights for management and decision making in the non-profit sector. In particular, this study fills four gaps in the current body of research, as follows:

(i)   This study improves the theories on value creation and cocreation in sports organizations. Specifically, value creation in Colombian soccer clubs takes place through new capabilities and new processes. The main common element in these clubs is the ability of individuals to create or improve products, services, or activities, which is in line with the findings reported in [69,70]. It is essential that professionals recognize the pivotal role of value creation in driving organizational success. Therefore, they should prioritize strategies that enhance customer/member value, sports performance, communication, and financial stability. This study is consistent with previous research, which emphasized the economic benefits that sports organizations can obtain through value creation. Moreover, practitioners should be aware that value-creation initiatives not only contribute to organizational performance, but also have the potential to attract favorable financial support, program participation, and physical outcomes in sports. By leveraging value creation for economic gain, NPSOs can enhance their financial viability and long-term sustainability.

(ii)  The study shows that value creation by sports clubs has a positive impact on the performance dimensions of soccer clubs, and that new services are sources of income, strategic alliances, improvements in organizational image, and more efficient administrative processes. All of this confirms previous findings in relation to the impact of value creation on the performance of sports organizations [1,5,9,45]. The findings highlight the mediating role of value cocreation in the relationship between value creation and organizational performance. This implies that organizations should actively involve customers/members, stakeholders, and other interested parties in their cocreation processes. By encouraging collaboration and shared decision making, NPSOs can improve their performance, tailoring strategies to address specific performance aspects. This study identifies various aspects that contribute to organizational performance, such as sport, customers/members, communication, image, finance, and organization. To optimize performance, organizations must assess their strengths and weaknesses in each aspect and develop targeted strategies to improve the areas that have the greatest impact on performance.

(iii) The existence of value cocreation in Colombian soccer clubs is demonstrated, which is a contribution to the body of research presented in [33,34]. Among other activities, these clubs hold formal and informal discussions for new service-design processes and to solve mutual problems using communication channels, as described in [26,27,71]. This study shows that the most positive effects of value-creation processes are on the customers/members of amateur soccer clubs. This highlights the importance of understanding and meeting customer/member needs in creative and innovative ways. By delivering value and building strong relationships with their target audience, NPSOs can attract new customers/members, ensure their loyalty, and receive ongoing support.

(iv)  This study proposes a conceptual model validated by the PLS-SEM method, thus contributing knowledge to the field of value creation and cocreation in Colombian soccer clubs, which are fragile and precarious sports organizations [21,22]. The study also provides valuable and unique information about the benefits of creating value in amateur soccer clubs in developing countries [72,73] to positively affect the dimensions of organizational performance. Furthermore, this study is one of the first attempts to

provide empirical evidence linking value creation, cocreation, and implementation in sports organizations in the South American context.

These practical implications provide guidelines for NPSOs seeking to improve performance, engage stakeholders, and drive innovation through value creation and cocreation processes. This study shows the importance of value-cocreation processes in achieving high levels of organizational performance. Therefore, sports and administration managers in NPSOs should intensify value cocreation in products, services, or activities to achieve results that favor performance in the sport, customers/members, communication and image, finance, and organization dimensions [62]. Thus, this study has implications for the managers of NPSOs in South America and other developing countries. Finally, the governing bodies of amateur soccer clubs around the world (e.g., FIFA) should recognize the importance of creating value together with partners, customers, sponsors, suppliers, and fans to build trusting relationships that lead to high levels of organizational performance.

*5.2. Limitations*

This study faced some limitations due to the few studies available on the relationships between value creation and cocreation and the performance of NPSOs in Colombia. The study was conducted in Antioquia, which is considered the region with the greatest level of soccer development in Colombia. Therefore, the results may differ from those of other regions of the country. In addition, the instrument used for the data collection was administered in Spanish, although the references were created in English. The scales and units of measurement used in the instrument were adapted for the amateur soccer clubs, so it is necessary to validate them in similar studies.

**Author Contributions:** Conceptualization, J.I.B.O., S.J.C.H. and L.C.H.C.; data curation L.C.H.C. and A.V.-A.; formal analysis, J.I.B.O. and L.C.H.C.; funding acquisition, S.J.C.H. and L.C.H.C.; investigation, J.I.B.O. and S.J.C.H.; methodology, J.I.B.O. and A.V.-A.; resources, J.I.B.O.; software, S.J.C.H. and L.C.H.C., supervision, J.I.B.O. and A.V.-A.; visualization, J.I.B.O. and S.J.C.H.; writing—original draft, J.I.B.O., S.J.C.H., L.C.H.C. and A.V.-A.; writing—review & editing, J.I.B.O., S.J.C.H., L.C.H.C. and A.V.-A. All authors have read and agreed to the published version of the manuscript.

**Funding:** This research received no specific grants from any funding agencies in the public, commercial, or not-for-profit sectors.

**Institutional Review Board Statement:** This study was conducted according to the guidelines of the Declaration of Helsinki and approved by the Institutional Review Board of Research Ethics Committees—ITM (protocol code: P17226. Date of approval: 13 September 2020).

**Informed Consent Statement:** Informed consent was obtained from all subjects involved in the study.

**Data Availability Statement:** The data may be provided free of charge to interested readers upon request through the corresponding author's email.

**Conflicts of Interest:** The authors declare no conflict of interest.

## Appendix A. Description of Constructs

**Table A1.** Equivalence of Original and Adapted Dimensions.

|  | First-Order Constructs | Original Dimensions | Adapted Dimensions |
|---|---|---|---|
| The author of [60] conducted research on value cocreation in Malaysian telecommunication companies. | Dialogue | Use different communication channels to hold dialogue sessions with consumers<br>Hold frequent dialogue sessions with consumers | The club uses different communication channels to engage in dialogue with consumers<br>The club frequently holds dialogue sessions with consumers |

**Table A1.** *Cont.*

| | First-Order Constructs | Original Dimensions | Adapted Dimensions |
|---|---|---|---|
| The author of [60] conducted research on value cocreation in Malaysian telecommunication companies. | Dialogue | Involve internal parties in dialogue sessions with consumers | The club involves its internal staff in dialogue sessions with consumers |
| | | Involve external parties in dialogue sessions with consumers | The club involves external entities in dialogue sessions with consumers |
| | | Recognize consumers' experiences with the service or product | The club recognizes consumers' experiences with its sports products |
| | | Emphasize employees' efforts in dealing with individual consumers | The club emphasizes employees' efforts in dealing with each consumer |
| | Access | Offer consumers the opportunity to participate in the service- or product-design process | The club offers consumers the opportunity to participate in the design process of their sports products |
| | | Offer consumers the opportunity to participate in the service- or product-development process | The club offers consumers the opportunity to participate in the process of making their sports products |
| | | Offer consumers the opportunity to participate in the service- or product-pricing process | The club offers consumers the opportunity to participate in the process of setting the price of their sports products |
| | Risk | Place more emphasis on providing consumer experiences than on the service or product ownership | The club emphasizes the delivery of consumer experiences based on the properties of its sports products |
| | | Provide consumers with all the necessary information related to the service or product | The club provides consumers with all the necessary information related to its sports products |
| | | Inform consumers of the potential risks of the service or product offered | The club informs consumers of the potential risks of the sports products offered |
| | | Inform consumers about the limitations and capabilities of the firm | The club informs consumers about its capabilities and limitations |
| | | Recognize the changing dynamics of consumer needs | The club recognizes the changing dynamics of consumer needs |
| | | Accept consumers' complaints about the service or product offered | The club accepts consumers' complaints about the sports products offered |
| | Transparency | Assume all risk-related responsibilities | The club assumes all the responsibility for the risks associated with their sports products |
| | | Provide consumers with clear information about the service or product | The club provides consumers with clear information about its sports products |
| | | Disclose price-related information to consumers | The club discloses the prices of its sports products to consumers |
| | | Benefit from information symmetry between consumers and the firm | The club benefits from the exchange of information with its consumers |
| | | Build trust among consumers through transparent information | The club builds consumer trust through transparent information |
| | | Provide consumers with up-to-date information | The club provides consumers with up-to-date information |
| The authors of [64] studied value creation in the manufacturing industry | New skills | Our employees receive ongoing training to develop new skills | The club's employees receive ongoing training to develop new skills |
| | | Our employees have up-to-date knowledge and skills compared to our direct competitors | The club's employees have up-to-date knowledge and skills compared to our direct competitors |
| | | We are constantly reflecting on which new skills are needed to adapt to changing market requirements | The club is constantly reflecting on the need for new skills to adapt to market changes |
| | New technologies | We keep our firm's technical resources up to date | The club keeps its technological resources up to date |
| | | Our technical equipment is very innovative compared to that of our competitors | The club's technological equipment is very innovative compared to that of its competitors |
| | | We regularly use new technology to expand our product-and-service portfolio | The club uses new technology to expand its product and service portfolio |

**Table A1.** *Cont.*

| | First-Order Constructs | Original Dimensions | Adapted Dimensions |
|---|---|---|---|
| The authors of [64] studied value creation in the manufacturing industry | New partners | We are constantly looking for new partners | The club is constantly looking for new business partners |
| | | We regularly take advantage of opportunities to integrate new partners into our processes | The club regularly takes advantage of opportunities to integrate new partners into its processes |
| | | We regularly evaluate the potential benefits of outsourcing | The club regularly evaluates the potential benefits of outsourcing |
| | | New partners regularly help us develop our business model | New partners regularly help strengthen the club's business model |
| | New process | We have recently made significant improvements to our internal processes | The club has recently made significant improvements to its internal processes |
| | | We implement innovative procedures and processes to manufacture our products | The club implements innovative processes to develop its sports products |
| | | We regularly assess our existing processes and make significant changes when necessary | The club regularly evaluates its existing processes and makes significant changes when necessary |
| The authors of [62] proposed a method to quantitatively assess organizational performance in the governing bodies of the French-speaking community | Elite sport | Obtain international sport results | The club seeks to obtain sports results (Sport_1) |
| | | Increase athletes' participation in international competitions | The club increases the participation of its athletes in international competitions (Sport_2) |
| | | Improve sports services for athletes | The club improves services for its athletes (Sport_3) |
| | | Increase sports activities for members | The club increases sports activities for its members (Sport_4) |
| | Customers | Preserve sporting values in society | The club preserves sporting values in society (Customer_M_1) |
| | | Improve the provision of non-sports services to members | The club improves the provision of non-sports services to its members (Customer_M_2) |
| | | Attract members | The club attracts new members (Customer_M_3) |
| | | Build members' loyalty | The club builds loyalty among its members (Customer_M_4) |
| | Communication and image | Promote a positive image of their sport in the media | The club promotes a positive image of soccer in the media (Com_Image_1) |
| | | Promote a positive image of their sport among members | The club promotes a positive image of soccer among its members (Com_Image_2) |
| | | Improve internal communication among members and clubs | The club improves internal communication with its members (Com_Image_3) |
| | | Improve tracking of internal communication with members | The club improves the tracking of internal communication with its members (Com_Image_4) |
| | Finance | Obtain financial resources | The club seeks financial resources (Finance_1) |
| | | Manage financial expenditure | The club manages its financial expenditure appropriately (Finance_2) |
| | | Manage self-financing capacity | The club manages its self-financing capacity (Finance_3) |
| | | Manage financial independence from the government | The club manages its financial independence from the government (Finance_4) |
| | Organization | Improve the administrative and sports staff's skills | The club improves its administrative and sports staff's skills (Organization_1) |
| | | Improve volunteer skills | The club improves volunteer skills (Organization_2) |
| | | Improve the headquarters' internal functioning | The club improves its headquarters' internal functioning (Organization_3) |
| | | Improve the headquarters' organizational climate | The club improves its headquarters' organizational climate (Organization_4) |

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
