# Peer review of "Creating Value in Non-Profit Sports Organizations: An Analysis of the DART Model and Its Performance Implications"

_ejihpe, doi:10.3390/ejihpe13090121_

Round 1

Reviewer 1 Report

This study examines the impact of value creation and co-creation of the DART model on the performance of non-profit sports organizations (NPSOs). The authors analyzed data from sports and administrative managers of NPSOs, specifically amateur soccer clubs in Colombia, using structural equation modeling. The results suggest that value creation positively influences the performance of NPSOs through co-creation of value. Despite limitations due to the novelty of investigating these relationships in the context of South America, Colombia, the findings contribute to our understanding of co-creation's mediating effect on performance dimensions in NPSOs in developing countries. The study highlights the benefits of co-creation for non-profit sports organizations and underscores the importance of value creation. Increasing research on the proposed constructs will enhance our understanding of innovation and its impact on NPSOs. Overall, in my opinions, I recommend accept the manuscript to publish in EJIHPE.

This study examines the impact of value creation and co-creation of the DART model on the performance of non-profit sports organizations (NPSOs). The authors collected data from sports and administrative managers of amateur soccer clubs in Colombia and used structural equation modeling (SEM) and partial least squares (PLS-SEM) for analysis.

The results indicate that value creation positively influences the performance of NPSOs through co-creation. This finding is significant, considering the context of South America and the novelty of studying the relationship between value creation and co-creation in NPSOs. The study sheds light on the mediating effect of co-creation on various performance dimensions such as sports, clients/members, communication and image, finances, and organization in developing countries.

The study also emphasizes the potential benefits of co-creation for non-profit sports organizations and underscores the importance of value creation in this context. It suggests that further research on the proposed constructs will deepen our understanding of innovation's impact on NPSOs.

Overall, this study provides valuable insights for managers and policymakers of non-profit sports organizations, particularly in developing countries, highlighting the importance of co-creation in enhancing performance. Given its contributions to the field and its relevance to practitioners, I recommend accepting the paper to publish in EJIHPE.

Reviewer 2 Report

Thank you for the authors' efforts in conducting this research. While I acknowledge the potential of the manuscript, I have identified some concerns which I will outline below:

1. I recommend spelling out the acronym (DART Model) in the title, abstract, and text whenever it is introduced for the first time, as not all readers may be familiar with the term. Additionally, the authors should provide an explanation of what the DART model is in the text.

2.    While the authors mentioned a previous research gap in line 6, it remains unclear what specific aspect has been missing in previous research and how this study aims to fill that gap. The authors should clearly articulate the research gap and its significance.

3.    In value creation, the authors identified four constructs. It is important to provide a rationale for how the authors adopted of these constructs. The same applies to value co-creation, where the authors should justify the selection of the four constructs. Furthermore, in performance, the authors introduced five constructs without providing a clear explanation of their relevance to performance. The authors should explain the reasoning behind identifying these five constructs and their connection to performance. as I’m not sure why those are related to performance.

4.    The research hypotheses were underdeveloped. The section did not adequately explain how and why the hypotheses were formulated. The authors should strengthen this section by providing stronger justification for each relationship proposed in the hypotheses.

5.    PLS-SEM is an acronym for Partial Least Squares Structural Equation Modeling. It is important to make this change in the abstract and include the full name when introducing it in the main body.

6.    While the authors did a good job of presenting value co-creation related literature, I’d further suggest incorporating recent sport management literature on the topic (see examples below). This will allow the authors further strength the section (i.e., 2.1.) and present up-to-date information to readers.

7.    While the authors presented a good review of literature related to value co-creation, I recommend incorporating recent sport management literature (examples below) on the topic to further enhance this section. This will strengthen the section (2.1) and provide up-to-date information to readers.

Kim, K., Byon, K. K., & Baek, W. (2020). Customer-to-customer value co-creation and co-destruction in sporting events. The Service Industries Journal, 40(9-10), 633-655.

Jiang, X., Kim, A., Kim, K., Yang, Q., García-Fernández, J., & Zhang, J. J. (2021). Motivational antecedents, value co-creation process, and behavioral consequences in participatory sport tourism. Sustainability, 13(17), 9916.

Kolyperas, D., Maglaras, G., & Sparks, L. (2019). Sport fans’ roles in value co-creation. European Sport Management Quarterly, 19(2), 201-220.

8.    Lastly, the implications section appears to be a repetition of the findings. It is essential to discuss how the findings can meaningfully advance our knowledge on the topic by addressing what the authors have discovered that previous research did not cover and how it can contribute to the field.

I hope these suggestions and comments will help improve the manuscript.

The writing quality of the manuscript needs significant improvement. There are several sentences that read awkwardly and are difficult to understand. For instance, in p. 2, line 64, the sentence "There is research on sports management in Colombia in a separate, non-compared way and without a rigorous and valid information system" is unclear due to the phrases "non-compared way" and "a rigorous and valid information system." The authors should enhance the readability of the entire manuscript by having it proofread by experts.

Author Response

June 30, 2023

Dear

EJIHPE – Editorial Team

Kind regards

In accordance with the suggestions of the reviewers in our article “Creating Value in Non-Profit Sports Organizations: An Analysis of the DART Model and its Performance Implications”, the following changes were made, properly marked with red letters in the article:

Section

Comment

Response

Title, abstract

I recommend spelling out the acronym (DART Model) in the title, abstract, and text whenever it is introduced for the first time, as not all readers may be familiar with the term. Additionally, the authors should provide an explanation of what the DART model is in the text.

The acronym was spelled out in the title, abstract, and text to provide clarity to the readers.

While the authors mentioned a previous research gap in line 6, it remains unclear what specific aspect has been missing in previous research and how this study aims to fill that gap. The authors should clearly articulate the research gap and its significance.

The existing gaps in the bibliography are specified and further elaborated upon.

Value Creation and Co-Creation

 In value creation, the authors identified four constructs. It is important to provide a rationale for how the authors adopted of these constructs. The same applies to value co-creation, where the authors should justify the selection of the four constructs. Furthermore, in performance, the authors introduced five constructs without providing a clear explanation of their relevance to performance. The authors should explain the reasoning behind identifying these five constructs and their connection to performance. as I’m not sure why those are related to performance.

The dimensions of Value creation and co-creation, and performance are exhaustived explained

Hypotheses and Conceptual Model

The research hypotheses were underdeveloped. The section did not adequately explain how and why the hypotheses were formulated. The authors should strengthen this section by providing stronger justification for each relationship proposed in the hypotheses.

The explanations of the origin and support of the hypotheses are included.

Abstract

 PLS-SEM is an acronym for Partial Least Squares Structural Equation Modeling. It is important to make this change in the abstract and include the full name when introducing it in the main body.

The acronym was spelled out  to provide clarity to the readers.

Value Creation and Co-Creation

While the authors did a good job of presenting value co-creation related literature, I’d further suggest incorporating recent sport management literature on the topic (see examples below). This will allow the authors further strength the section (i.e., 2.1.) and present up-to-date information to readers.

The literature was strengthened with the suggestions from the reviewer.

Value Creation and Co-Creation

While the authors presented a good review of literature related to value co-creation, I recommend incorporating recent sport management literature (examples below) on the topic to further enhance this section. This will strengthen the section (2.1) and provide up-to-date information to readers.
Kim, K., Byon, K. K., & Baek, W. (2020). Customer-to-customer value co-creation and co-destruction in sporting events. The Service Industries Journal, 40(9-10), 633-655.
Jiang, X., Kim, A., Kim, K., Yang, Q., García-Fernández, J., & Zhang, J. J. (2021). Motivational antecedents, value co-creation process, and behavioral consequences in participatory sport tourism. Sustainability, 13(17), 9916.
Kolyperas, D., Maglaras, G., & Sparks, L. (2019). Sport fans’ roles in value co-creation. European Sport Management Quarterly, 19(2), 201-220.

The literature was strengthened with the suggestions from the reviewer.

 Implications

Lastly, the implications section appears to be a repetition of the findings. It is essential to discuss how the findings can meaningfully advance our knowledge on the topic by addressing what the authors have discovered that previous research did not cover and how it can contribute to the field.

The implications were further explored by addressing the results and how it can contribute to the field.

We look forward to your comments and hope to hear from you soon.

Thank you very much

_

The authors

Round 2

Reviewer 2 Report

Thank you for the revision.

Good